# FAIR-SP: CAPTURING PLURALISTIC SOCIAL EQUITY PREFERENCES THROUGH SYNTHETIC DATA

## ABSTRACT

Human preference plays a crucial role in understanding social values and developing inclusive AI systems. However, collecting comprehensive human preference feedback is costly, and most existing datasets neglect the pluralism of social segment preferences, particularly in social equity domains. To address this gap, we introduce FAIR-SP, a synthetic dataset capturing pluralistic social segment preferences on equity issues, systematically constructed with theoretical guidance from multiple disciplines including sociology and philosophy. FAIR-SP encompasses 28 social groups, 98 equity topics, and 5 preference dimensions. Through automatic question generation mechanisms, it provides both concise template-based and narrative-driven contextualized scenario questions, yielding 238,623 preference records via GPT-4o-mini role-playing based on seven representative UK public segments, with extensions to other regional contexts. We validate the dataset quality through multiple complementary approaches, achieving over 90% role-play fidelity and human evaluation scores exceeding 0.7. We demonstrate the dataset utility through targeted equity preference alignment experiments and equity positioning analysis of LLMs across global regions. FAIR-SP establishes a foundational resource for understanding and incorporating pluralistic social values especially in the era of LLMs.

## 1 INTRODUCTION

With the growing adoption of LLMs in public policy making and public services[1234], a key question is *How can an LLM-based public policy maker accurately capture and represent pluralistic and dynamic public segment preferences of social values?* On one hand, social values like attitudes towards equity can vary significantly among different social segments (Huseman et al., 1987; Surridge, 2021; Tuli et al., 2023). On the other hand, to promote the alignment of LLMs with pluralistic societal values is crucial to achieve societal safety (Ji et al., 2023; Qi et al., 2024; Yin et al., 2024; Huang et al., 2024), support cultural inclusivity (Tao et al., 2024; Alkhamissi et al., 2024; Li et al., 2024a;b), and also reflect pluralistic human values (Durmus et al., 2023; Santurkar et al., 2023; Sorensen et al., 2024; Zhao et al., 2024). For the alignment, high-quality human feedback of their social-value preferences is necessary. However, its collection can incur significant costs (Dubois et al., 2023; Cui et al., 2023). Moreover, social value preferences may evolve over time, influenced by factors such as demographic shifts and social development (Greenfield, 2016; Ramos et al., 2019; Zarwi et al., 2017).

Despite increasing efforts in exploring and collecting human preference data of social values, existing datasets face several limitations: (i) **Social equity remains unaddressed**: though an important issue in social psychology, it is considered little[5] (ii) **Neglecting pluralism of value preferences of different public segments**: Universal viewpoints *e.g.*, cultural and political beliefs (Li et al., 2024b) receive significant attention while pluralism of segment value preferences remains largely unexplored (Huseman et al., 1987; King Jr & Miles, 1994).

---

[1]https://openai.com/global-affairs/introducing-chatgpt-gov/

[2]https://openai.com/index/democratic-inputs-to-ai-grant-program-update/

[3]https://huggingface.co/stewhsource/GovernmentGPT

[4]https://en.wikipedia.org/wiki/Diella_(AI_system)

[5]https://www.un.org/en/actnow/ten-actions-just-society

(iii) **Single dimension employed**: insufficient attention is given to the multifaceted nature of segment value preferences spanning multiple dimensions. (iv) **Insufficient guidance from social science**: typically data are generated using information sources from computer science domain or by manually hand crafting, and the empirical foundations and insights from sociological domain is overlooked.

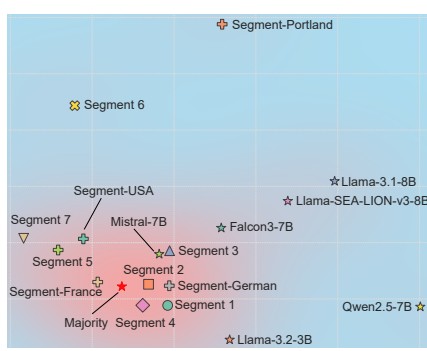

Figure 1: Landscape of FAIR-SP space, closer positions denote more similar equity preferences (Appendix I for detail.)

We release FAIR-SP, a human social-value preferences resource capturing fine-grained pluralistic equity preferences of different public segments. First, we develop a comprehensive question bank consisting of 34,089 generated multi-choice questions asking about equity perspectives. Both concise template-based questions and narrative-driven contextualized scenario-based questions are included, enabling comprehensive capturing of nuanced group-specific conceptualizations of equity across various social scenarios. The question bank systematically integrates established categorizations from sociology and philosophy domain, comprising: ① 28 social groups of equity-affected populations; ② 98 equity topics expanded from The Fairness Foundation's categorization of fair necessities[6]; and ③ five equity dimensions collected from equity study in multiple disciplines, each capturing contrasting perspective on equity conceptualization and prioritization. A generated question example is shown on the right side of Figure 9[7]. Then, we employ GPT-4o-mini with a role-playing mechanism to simulate different public segments, eliciting segment-specific equity preferences responses. Our mechanism allows to use public segmentation from different regions. W.l.o.g, we mainly use the segmentation of the UK public (including seven segments, derived based on a real social quiz) as reported in Britain's Choice (Surridge, 2021) for analysis and presentation, yielding 238,623 segment preference data points[8]. Figure 9 presents an overview of the construction of FAIR-SP.

We validate quality of FAIR-SP through multiple complementary approaches. First, role-play fidelity is assessed by having each segment-assigned LLM complete the quiz from used for the segmentation, achieving over 90% classification accuracy across all the segments (Figure 5). Second, we employed 20 human annotators to evaluate whether the generated preference data conform with intended segment characteristics, achieving average scores exceeding 0.70 (Table 4). The human evaluation not only further validates role-playing fidelity, but also indirectly confirms the interpretability (comprehensible and meaningful equity scenarios) and discriminant validity of our question bank, as the annotators could consistently distinguish different segments' equity perspectives from the QA pairs.

We further demonstrate utility of FAIR-SP for targeted segment preference alignment via model fine-tuning, where multiple approaches are considered[9], and the results show that its introduce significantly improved alignment scores for all these approaches compared to a pure prompt-based strategy. Further evaluation with the real-world segmentation quiz also confirms the alignment performance.

Our contributions can be summarized as follows:

1. We introduce FAIR-SP, which to the best of our knowledge, is the first social value preference dataset specialized in equity. Its construction is guided by the established conceptualization and categorization of equity from multiple disciplinary.

2. We propose flexible, automatic question generation and role-play mechanisms, which allows extension or update of equity topics and perspectives, social groups in concern, and also public segmentation.

---

[6]A social research charity focused on promoting social equity (Snell, 2021)

[7]More examples are in Appendix D.

[8]Results using public segmentation of other regions are in Appendix H

[9]Besides SFT and DPO, we propose a new sample reweighting approach to emphasize samples exhibiting segment uniqueness and it demonstrates superior performance.

3. FAIR-SP facilitates not only LLM alignment w.r.t. pluralistic value preferences of social segments, but also comprehensive comparative studies of social value positions of constantly emerging and updated LLMs from different global regions.

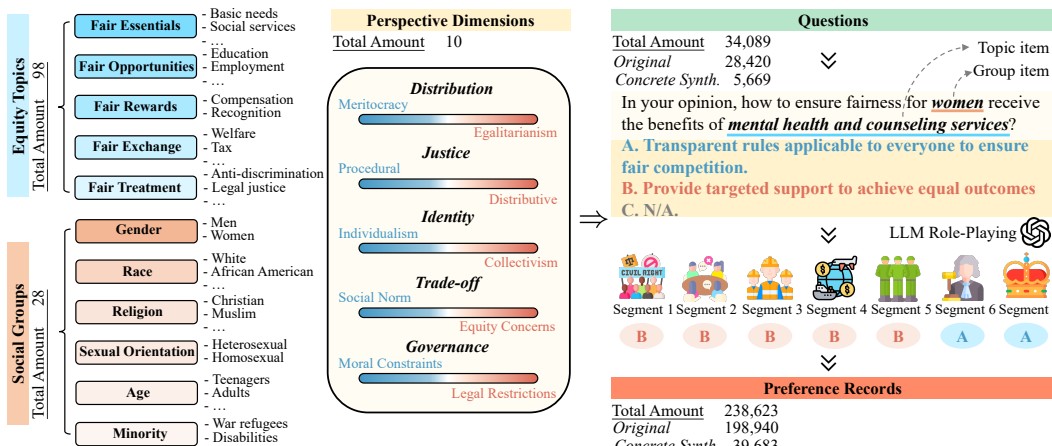

Figure 2: An overview of the FAIR-SP dataset. Each question consists of three parts: the social group, an equity topic, and a perspective dimension. An example question is shown on the right where option A and B represent two types of viewpoints under a specific dimension. Social segment preferences are collected through LLM segmentlization that leverages 7 value portrait based on the real-world social surveys.

## 2 FAIR-SP

FAIR-SP is a multi-level resource for segment equity preferences, features questions structured around three key components: social groups, equity topics, and perspective dimensions, which are further detailed in Section 2.1–2.3. The methodology for question generation is subsequently introduced in Section 2.4. Subsequently, we present the segment preferences captured through LLM-segmentlization in Section 2.5.

### 2.1 SOCIAL GROUPS

FAIR-SP covers a total of 28 social groups, including common standard social group categories like gender, age, race, religion, and sexual orientation, as well as a range of typical social minority groups such as 911 victims, Black Lives Matter supporters, war refugees, famine victims, feminists, and environmentalists. These selections reflect representative social concerns and historical contexts from various regions worldwide. More details are presented in Appendix B.

### 2.2 SOCIAL EQUITY TOPICS

To integrate real-world equity preferences during dataset construction, FAIR-SP draws upon five major equity topics identified through surveys conducted by The Fairness Foundation, a real-world social research organization. Building on these, we consider a comprehensive set of subtopics including basic living needs, healthcare, education, employment, finance, law, and other relevant social issues. An overview is provided in Figure 9, with the specific categories listed as follows:

**Fair Essentials**. Meeting people's basic needs is fundamental to achieving social equity. Within the concept of fair essentials, we identify four fundamental needs: (1) Basic material needs: this encompasses the essentials for survival and well-being, such as food, clean water, and shelter. (2) Basic health needs: access to essential medications, basic sanitation, and healthcare services are crucial for maintaining health. (3) Basic social services: everyone deserves to feel safe, have access to public transportation, and receive the basic education, enabling them to participate fully in society. (4) Fundamental rights: human rights, freedom of speech, and other fundamental freedoms are essential for individual autonomy and dignity.

**Fair Opportunities**. Everyone deserves the chance to achieve success in life. We categorize fair opportunities into three key areas: (1) Education and skills development: access to affordable higher education, vocational training, and lifelong learning opportunities empowers individuals to gain the knowledge and skills needed to thrive. (2) Economic and employment: this encompasses fair access to jobs, opportunities for advancement, and career switch, ensuring that everyone has the chance to contribute to the economy and achieve economic security. (3) Political participation: including exercising right to vote, running for office, and policy feedback access, which offers avenues for citizens to engage in public policy.

**Fair Rewards**. This principle emphasizes that everyone should be justly rewarded for their efforts and contributions. To explore this concept, we identify two main categories: (1) Compensation: including wages, bonus and tips, which focuses on physical rewards within the workplace. (2) Social recognition: recognizing and appreciating individual efforts publicly, such as verbal praise and media shout-outs.

**Fair Exchange**. Aims to ensure a balance between individuals' social welfare and tax payments, which can be broadly categorized into three main areas: (1) Reciprocity: focuses on providing support and benefits to individuals, such as unemployment benefits and disability supports. (2) Welfare: encompasses a range of services and programs designed to improve the well-being of individuals and families, including subsidized childcare, free legal aid, and mental health counseling. (3) Tax: various taxes levied on individuals and businesses to fund social welfare programs and public services, typically with income tax, inheritance tax and luxury tax.

**Fair Treatment**. Fair Treatment ensures that people are treated equitably in all aspects of society. For this topic, we categorize three key themes: (1) Anti-discrimination: this includes protection against stigmatization, culturally inclusive healthcare services, and more. (2) Legal and social justice: This encompasses protection from workplace harassment, safeguards against exploitative contracts, and other measures. (3) Public resource equity: which involves initiatives such as the distribution of public restrooms in underserved areas, and support for public housing programs.

For each subtopic, we further divide it into more specific subject matters, eventually resulting in a total of 97 specific topics. For details, please refer to Appendix B.

## 2.3 SOCIAL SEGMENT PREFERENCE DIMENSIONS

In designing the five preference dimensions, we draw on sociology, political theory, ethics, and cultural psychology[10]. This includes core conceptions of social equity, such as distributive ideology (Dimension 1), justice theory (Dimension 2), and political ideology (Dimension 3), alongside complementary perspectives captured by social theory (Dimension 4) and governance philosophy (Dimension 5). Each dimension includes two distinct orientations. Specifically,

Dimension 1 (**Meritocracy** *vs.* **Egalitarianism**) (Goto, 2022): *Should we prioritize current achievements or promoting evenly outcomes?*

This dimension contrasts two approaches to equity: meritocracy, where rewards are based on achievements (*e.g.*, promotions based on performance), and egalitarianism, which emphasizes even outcomes (*e.g.*, distributing resources evenly). The debate centers on whether merit or equality should be prioritized in equity judgments.

Dimension 2 (**Procedural** *vs.* **Distributive**) (Clay-Warner et al., 2005): *Should justice emphasize fair competitive or prioritize supporting the disadvantaged to achieve equal outcomes?*

Dimension 2 highlights the contrast between two conceptions of equity: procedural justice (fair processes, e.g., decisions based on neutral rules like standardized testing) and distributive justice (fair outcomes, e.g., corrective policies like affirmative action). The core issue is whether equity depends on impartial procedures or equitable results.

Dimension 3 (**Individualism** *vs.* **Collectivism**) (LeFebvre & Franke, 2013): *Should resources be shared based on individual efforts or collective allocation?*

This dimension addresses whether equity should emphasize individual responsibility and effort or prioritize collective well-being by emphasizing social responsibility, highlighting differing perspectives on how equity is understood either through segmental contribution or through shared obligations and group-oriented outcomes.

---

[10]More details are in Appendix C

Dimension 4 (**Social norm** *vs.* **Equity concerns**) (Bussolo et al., 2024): *Should we prioritize adherence to established social norms or the pursuit of equity?*

This dimension contrasts social norms (*e.g.*, maintaining traditional gender roles) with equity concerns (*e.g.*, advocating for gender equality in the workplace). The question is whether to preserve tradition or to promote equity, even if it challenges societal conventions.

Dimension 5 (**Moral** *vs.* **Law**) (Alder & Gilbert, 2006): *Should equity be achieved primarily through moral constraints or legal constraints?*

This dimension examines whether equity should be guided primarily by moral principles or by legal constraints. It addresses the question of whether ethical considerations or formal legal frameworks ought to serve as the foundation for fair treatment.

### 2.4 QUESTION DATA GENERATION

**Concise template-based.** We created a multiple-choice questionnaire with a total of 28,420 samples, where questions combined social groups, equity topics, and perspective dimensions, as described in the sections above. Corresponding to preference dimensions, each question includes three options: two opposing viewpoints and an N/A option to avoid bias due to forced selections.

**Narrative-driven contextualized scenario-based.** Furthermore, to improve the diversity of the data, we sample 5,669 questions from the dataset and generate concrete scenario samples using GPT-4o. For each question and its corresponding options, we prompt the model to create short real-world scenarios that reflect each perspective. Each scenario includes the background of a fictional person, a decision point where they receive a service or resource, and a brief emotional response. This variant process yields more realistic and pluralistic scenarios, enabling the in-depth analysis of segment preferences across a variety of situations. For more details, please refer to Appendix D, which includes the detailed prompts and an example data point presented in Figure 6.

### 2.5 SOCIAL SEGMENT PREFERENCE

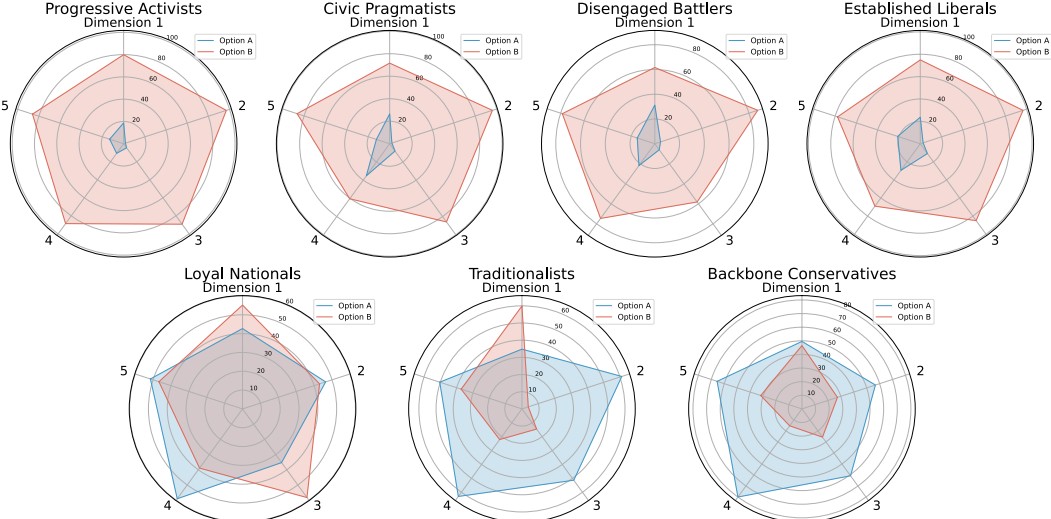

Figure 3: Social segment preference anchors. Blue and red represent the proportions of choices for option A and option B, respectively.

**Segments.** Drawing on the social segment typologies derived from real-world surveys conducted by More in Common, which identify seven segments in UK society[11], we anonymize the country-related descriptions (*e.g.,* preferred media outlets and supported political parties) to construct universal segment profiles. Specifically, the seven segments are as follows[12]: **Segment 1** (*Progressive Activists*),**Segment 2** (*Civic Pragmatists*), **Segment 3** (*Disengaged Battlers*), **Segment 4** (*Established Liberals*), **Segment 5** (*Loyal Nationals*), **Segment 6** (*Disengaged Traditionalists*), **Segment 7** (*Backbone Conservatives*).

---

[11]More in Common is a widely recognized social research organization in the UK (Juan-Torres et al., 2020).

[12]The detailed description are in Appendix E

For more detailed segment prompts, please refer to Appendix E. Considering regional differences, we also provide the FAIR-SP-CN dataset in the Chinese context in Appendix H.1 and cross-regional segment mappings in Appendix H.2.

**Social segment preference.** Given that advanced LLMs demonstrate strong role-play capabilities, we leverage GPT-4o-mini (Achiam et al., 2023) to simulate seven representative segments to capture the pluralistic segment preference on the questions discussed in Section 2.4, complemented by self-calibration prompt (Li et al., 2024b) to further enhance consistency.

We then analyze the similarities and differences in the preferences of these seven segments, based on the responses generated by respective models, as shown in Figure 3, which illustrates the choice preferences of seven segments across five perspective dimensions. For instance, Segment 1 (Progressive Activists) shows a greater preference for option B (*e.g.*, equal outcomes), whereas Segment 5 (Loyal Nationals) exhibits a more balanced preference distribution, and Segment 7 (Backbone Conservatives) indicates a preference towards option A (*e.g.*, fair competitive and prioritize social norms). Note that the preference space is continuous, making exhaustive enumeration of all segmentlity preference types fundamentally intractable. Despite that, based on responses from seven representative segments, we establish these preference profiles as anchor points, which are subsequently used to position new test points within the segment preference space.

## 2.6 DATASET ANALYSIS AND VALIDATION

**FAIR-SP captures the diversity of social segment preferences.** We present a detailed analysis of the segment preference data. As shown in Figure 4, which presents a fine-grained view of the seven segments' differential preferences for options when considering equity topics within various social groups (bottom scatter plot), and the aggregate distribution of all votes across these dimensions by all segments (left and top bar plot). We then examine its commonalities and differences across segments, which reveal the fundamental structure of preference patterns within the dataset, offering a descriptive overview of its key attributes and establishing a basis for subsequent research into potential influencing factors. For more detailed analysis, please refer to Appendix F.

**Role-play fidelity validation.** We validated LLM role-play fidelity by having each segment-assigned LLM complete the More in Common segmentation quiz (Juan-Torres et al., 2020) designed for human public and measuring whether the quiz correctly classified the LLM into its intended segment (refer to Appendix G.3 for more details). As shown in Figure 5, for all the segments over 90% fidelity rate is achieved, confirming the effectiveness of our role-play design.

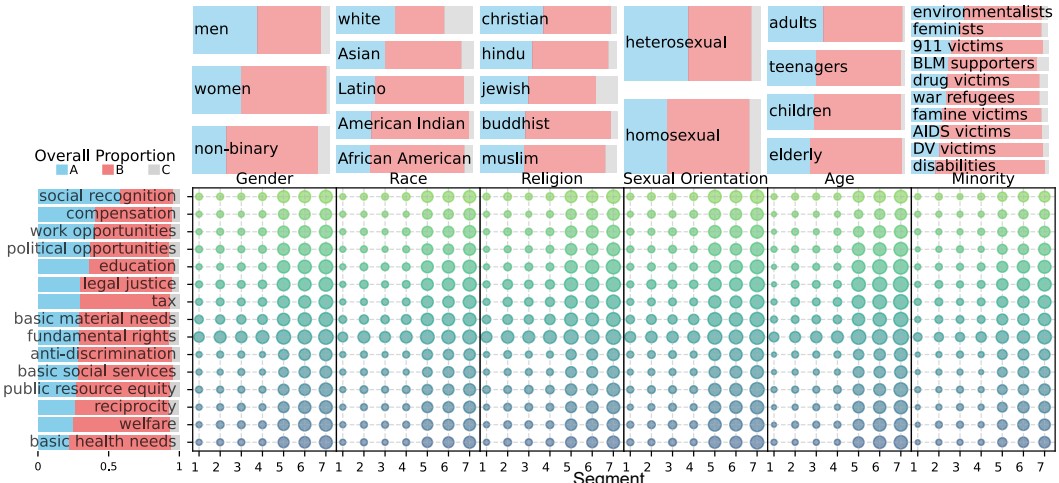

Figure 4: Fine-grained segment preferences and aggregate distribution across segments, social groups, and equity topics: The scatter plot shows the proportion of option A selected for each segment across the combined social group and equity topic categories (with point size scaled by the proportion). The bar plots on the left and top show the overall option distribution for each equity topic and social group, respectively.

**Human evaluation of preference data.** Besides letting role-play LLM directly take segmentation quiz, we also validated whether the preference data generated by them conform to their intended segments, as perceived by real humans (details are provided in Appendix G.4). This reflects whether our question bank for capturing equity preferences is reasonable for human evalutors. As shown in Table 4, the average score exceed 0.70, confirming our generated preference data is consistent with segments to an obvious extent, as perceived by real human annotators.

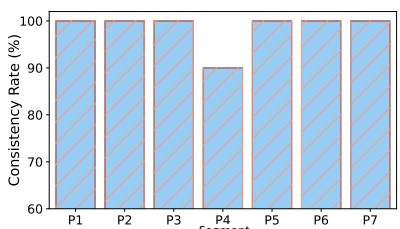

Figure 5: Role play fidelity of seven segment LLMs on real-world quiz.

# 3 FAIR-SP FOR LLM ANALYSIS AND ALIGNMENT

## 3.1 MAINSTREAM LLMS EQUITY PREFERENCE DISTRIBUTION

**Setup.** We choose six representative models from different regions to explore segment preference, including Falcon3-7B-Instruct (Arab) (Team, 2024), Llama-3.1-8B-Instruct, Llama-3.2-3B-Instruct (North America) (Grattafiori et al., 2024), Llama3.1-8b-cpt-sea-lionv3-instruct (Southeast Asia) (Ng et al., 2025), Mistral-7B-Instruct-v0.3 (Europe) (Jiang et al., 2023), and Qwen2.5-7B-Instruct (China) (Yang et al., 2024). Following (Feng et al., 2024), we calculate the 1 - Jensen-Shannon distance between each model and each segment. For detailed experimental setup, please refer to the Appendix G.1.

**Results.** The results are shown in Table 1, although the distribution of JS Distance differs between models, all models consistently show the closest similarity to Segment 3. Especially, Qwen2.5-7B, Mistral-7B, and Falcon3-7B demonstrate high similarity scores of 0.98, 0.94, and 0.91 respectively, the remaining three Llama-based models show relatively lower values. Generally, all the models are closer to the first four segments and relatively farther from the last three segments. We also provide results on FAIR-SP-CN in Appendix I.2.

## 3.2 ALIGNMENT TO TARGETED SEGMENT

### 3.2.1 REWEIGHTING

FAIR-SP offers a clear observation of how preferences vary between different segments. Unlike existing preference datasets and alignment techniques that steer optimization toward the majority preference, which ignore the pluralistic distribution of viewpoints, FAIR-SP recognizes the diversity of preferences and offers a practical path toward segment preference optimization. Intuitively, commonalities reflect the collective preferences of the public, while differences highlight the uniqueness of each segment. Rather than treating all samples uniformly, we reweight samples to emphasize those exhibiting segment-specific uniqueness. Formally, the weight for each is as follows:

$$W_i = \frac{T_i/N_i}{\sum_{j=1}^{K} T_j/N_j} \tag{1}$$

where $T_i$ denotes the index of frequency tier, $N_i$ is the sample count for tier $i$, and $K$ is the number of frequency tiers. Given that preference frequency tiers vary across segments, sample weights differ accordingly for each target segment. More details please refer to Appendix G.2.

Table 1: The Jensen-Shannon similarity between representative models from different regions and preference anchor points. The underlined values represent the nearest segment for each model.

|  | 1 - Jensen-Shannon Distance | | | | | | |
|---|---|---|---|---|---|---|---|
|  | S1 | S2 | S3 | S4 | S5 | S6 | S7 |
| Falcon3-7B | 0.79 | 0.80 | 0.91 | 0.78 | 0.68 | 0.64 | 0.60 |
| Llama-3.2-3B | 0.79 | 0.79 | 0.88 | 0.75 | 0.64 | 0.60 | 0.55 |
| Llama-3.1-8B | 0.53 | 0.55 | 0.66 | 0.53 | 0.53 | 0.62 | 0.52 |
| Llama3.1-8B-sea-lionv3 | 0.65 | 0.66 | 0.77 | 0.64 | 0.60 | 0.64 | 0.56 |
| Mistral-7B | 0.87 | 0.92 | 0.94 | 0.90 | 0.78 | 0.66 | 0.66 |
| Qwen2.5-7B | 0.85 | 0.88 | 0.98 | 0.85 | 0.74 | 0.65 | 0.63 |

Table 2: Performance comparison of different alignment methods on testing data for Llama-3.2-3B-instruct: * indicates the alignment target. The underlined values represent the nearest segment for each model, while bold values highlight the best-performing models targeting each segment.

| | | | 1 - Jensen-Shannon Distance | | | | |
|---|---|---|---|---|---|---|---|
| | S1↓ | S2↓ | S3↓ | S4↓ | S5↓ | S6 (*)↑ | S7↓ |
| Unaligned, Vanilla | 0.80 | 0.80 | 0.89 | 0.78 | **0.65** | 0.60 | **0.57** |
| Unaligned, Role Play | 0.80 | 0.87 | 0.94 | 0.84 | 0.80 | 0.72 | 0.71 |
| Aligned, SFT | 0.58 | 0.65 | 0.72 | 0.63 | 0.80 | 0.94 | 0.84 |
| Aligned, DPO | 0.56 | 0.63 | 0.69 | 0.62 | 0.81 | **0.98** | 0.89 |
| Aligned, WSFT | 0.54 | 0.61 | 0.68 | 0.59 | 0.77 | 0.97 | 0.84 |
| Aligned, WDPO | **0.52** | **0.59** | **0.64** | **0.57** | 0.77 | **0.98** | 0.86 |

Table 3: Performance Comparison of Different Alignment Methods on Simulation Data for Llama-3.2-3B-instruct.

| | | | 1 - JS Distance | | | | |
|---|---|---|---|---|---|---|---|
| | S1↓ | S2↓ | S3↓ | S4↓ | S5↓ | S6 (*)↑ | S7↓ |
| Unaligned, Vanilla | 0.73 | 0.77 | 0.72 | 0.79 | 0.73 | 0.61 | **0.67** |
| Unaligned, Role-play | 0.53 | 0.58 | 0.57 | 0.59 | 0.93 | 0.84 | 0.90 |
| Aligned, SFT | 0.55 | 0.60 | 0.63 | 0.61 | 0.87 | 0.79 | 0.84 |
| Aligned, DPO | 0.60 | 0.65 | 0.62 | 0.66 | 0.88 | 0.77 | 0.82 |
| Aligned, WSFT | 0.52 | 0.57 | 0.52 | 0.59 | 0.82 | **0.87** | 0.83 |
| Aligned, WDPO | **0.28** | **0.32** | **0.35** | **0.33** | **0.70** | 0.82 | 0.76 |

### 3.2.2 ALIGNMENT EVALUATION WITH TEMPLATE DATA

**Setup.** We randomly split the FAIR-SP dataset into an 80% training set and a 20% test set. Then we select Segment 6 as alignment target, compare the performance of role-play, supervised fine-tuning (SFT), direct preference optimization (DPO) (Rafailov et al., 2023), SFT with sample reweighting (WSFT), and DPO with sample reweighting (WDPO) on the Llama-3.2-3B-instruct model. For role-play, we use the same role prompt as the target segment, please refer to Appendix E for details.

**Results.** As shown in Table 2, a straightforward prompt-based role-playing strategy fails to adequately achieve alignment, achieve 0.60 on segment 6 but 0.89 on Segment 3. Conversely, the implementation of SFT and DPO specifically targeting the desired segment demonstrates better results, achieve 0.98 and 0.94 on Segment 6 respectively. However, SFT and DPO still lack the ability to more precisely capture a segment's uniqueness, *i.e.*, to maximize the distance from other segments while aligning with the target. Applying sample reweighting to the training data effectively addresses this problem, as demonstrated by WSFT and WDPO in the results, these methods achieved high alignment scores of 0.97 and 0.98 towards segment 6, respectively, while simultaneously increasing the margin from other segments by 10.20% and 12.00% compared to vanilla. See the Appendix I.3 for more results on alignment targeting Segment 1.

### 3.2.3 ALIGNMENT EVALUATION WITH SIMULATION DATA

**Setup.** We further conduct experiments on the generation-based simulation data from Section 2.4 to assess the generalization performance of different alignment methods. The fine-tuned models are identical to those in Section 3.2 which were trained on the FAIR-SP training data.

**Results.** As shown in Table 3, while Role-play, SFT, and DPO are able to increase the similarity with Segment 6 on the simulation data, gain 0.84, 0.79 and 0.77, they fail to effectively reduce the distance from Segment 5 and Segment 7, *e.g.,* all these methods incorrectly aligned to Segment 5. In contrast, WDPO achieves alignment with Segment 6 with a highest score 0.87, while maximizing the differentiation from other segments, decrease 37.80% compared to vanilla.

Table 4: Human evaluation score.   Table 5: Segment quiz: Vanilla, Role-play, and WDPO models.

| | Average Score |
|---|---|
| Yes/No | 0.70 |
| Comparative | 0.72 |
| Overall | 0.72 |

| | S1 | S2 | S3 | S4 | S5 | S6 (*)↑ | S7 |
|---|---|---|---|---|---|---|---|
| Vanilla | 0 | 0 | 6 | 4 | 0 | 0 | 0 |
| Role-play | 0 | 0 | 2 | 0 | 0 | 5 | 3 |
| WDPO | 0 | 0 | 0 | 0 | 0 | **10** | 0 |

### 3.2.4 ALIGNMENT EVALUATION WITH THE REAL-WORLD QUIZ

We further validated via real-world quiz (trials provided by humans, refer to More in Common) whether the models are aligned to the target segment using our generated preference dataset. We tested Llama-3.2-3B vanilla, role-play, and WDPO models across 10 quiz trials each. Results in Table 5 show the vanilla model classified as Segment 3 (6/10 trials) and Segment 4 (4/10 trials). Role-play correctly identified as Segment 6 in only 5 trials, while WDPO consistently matched the target segment in all 10 trials, demonstrating reliable and stable alignment.

## 4 RELATED WORK

### 4.1 HUMAN PREFERENCE BENCHMARKS AND DATASETS

Recently, an increasing number of studies have focused on improving pluralistic representation and enhancing the alignment of Large Language Models (LLMs) with human preference (Christiano et al., 2017; Ziegler et al., 2019; Bai et al., 2022). As datasets focusing on subjective opinions, OpinionQA (Santurkar et al., 2023) and GlobalOpinionQA (Durmus et al., 2023) reveal a notable misalignment between the perspectives reflected by LLMs and those of different demographic groups. (Feng et al., 2024) introduce the ValuePrism dataset, which helps LLMs better capture pluralistic human values and reduce the underrepresentation of minority perspectives. Furthermore, (Rao et al., 2024) present NORMAD, a framework and dataset for evaluating the cultural adaptability of LLMs. (Li et al., 2024a) utilize augmented data derived from the World Values Survey (WVS) to introduce cultural diversity into LLMs. Building on this, CulturePark (Li et al., 2024b) leverages a multi-agent communication framework to generate richer cultural data for fine-tuning culture-specific models. Through enhanced analysis of data composition, PRISM (Kirk et al., 2024) delivers more culturally pluralistic preference data.

### 4.2 HUMAN PREFERENCE ALIGNMENT

Reinforcement Learning from Human Feedback (RLHF) has become a key method for aligning LLMs with human preferences. (Rafailov et al., 2023) introduce direct preference optimization (DPO), simplifying the preference tuning process by enabling direct policy optimization with a simple classification loss. (Ramesh et al., 2024) propose Group Robust Preference Optimization (GRPO) to improve alignment by optimizing for worst-case group performance. (Wang et al., 2024) introduce a Mixture-of-Experts (MoE) reward model enabling interpretable, multi-objective preferences. (Balepur et al., 2025) develop a two-stage framework for segment-based segmentlization. (Feng et al., 2024) promote pluralistic alignment through collaboration between a base LLM and public-specific models. These approaches demonstrate the growing trend toward more refined, context-aware, group-sensitive, and segment alignment strategies.

## 5 LIMITATIONS AND CONCLUSIONS

This paper presents FAIR-SP, the first synthetic dataset capturing pluralistic social segment preferences on equity issues across multiple social groups, equity topics, and preference dimensions. Through rigorous validation and systematic analysis of mainstream LLMs across global regions, we demonstrate consistent equity orientation patterns and propose an effective sample reweighting alignment method. FAIR-SP establishes a foundational resource for developing more inclusive AI systems, advancing social responsibility in the era of widespread LLM deployment. FAIR-SP has following limitations we are working on: (1) social survey with human participants from pluralistic public segments could further help validate and enhance our data quality. (2) our differential weighting approach currently focuses on individual segments, extending this to group-level analyses helps to capture both shared and divergent preferences within a society, providing more convenience for LLM-based policy makers.

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

## A  LLM USAGE

Large Language Models (LLMs) were used to aid in the writing and polishing of the manuscript. It is important to note that the LLM was not involved in the ideation, research methodology, or experimental design. All research concepts, ideas, and analyses were developed and conducted by the authors.

# B  DETAILS ABOUT SOCIAL GROUPS AND EQUITY TOPICS

We provide the detailed information of social groups as follows,

## SOCIAL GROUPS

- **Gender**
  - Men
  - Women
  - Non-binary
- **Race**
  - White
  - Asian
  - African American
  - American Indian
  - Latino
- **Religion**
  - Christian
  - Buddhist
  - Hindu
  - Jewish
  - Muslim
- **Sexual Orientation**
  - Heterosexual
  - Homosexual
- **Age**
  - Children
  - Teenagers
  - Adults
  - Elderly
- **Minority**
  - 911 victims
  - AIDS victims
  - Domestic violence victims
  - Drug victims
  - War refugees
  - Famine victims
  - People with disabilities
  - Black Lives Matter supporters
  - Feminists
  - Environmentalists

and the comprehensive list of equity topics is presented below,

- **Fair Essentials**
  - Basic Material Needs
    * Food
    * Clean water
    * Energy
    * Warm clothing

          * Stable shelter
          * Proper toilets
      – Basic Health Needs
          * Accessible basic healthcare
          * Basic sanitation facilities
          * Routine vaccinations
          * Public health services
          * Essential medications
          * Emergency medical services
      – Basic Social Services
          * Ensured public security
          * Ensured segmentl safety
          * Fire and rescue services
          * Quality primary education
          * Accessible public transport
          * Reliable waste disposal services
          * Affordable communication
      – Fundamental Rights
          * Basic law enforcement
          * Protected fundamental human rights
          * Right to liberty
          * Guaranteed freedom of speech
          * Guaranteed freedom to move
          * Guaranteed right to own property
  • **Fair Opportunities**
      – Education
          * Affordable higher education
          * Accessible vocational training
          * Accessible lifelong learning
          * Scholarship opportunities
          * Access to digital literacy
          * Effective career guidance
      – Employment
          * Job access
          * Promotion opportunities
          * Capital access
          * Training grant chance
          * Business loan access
          * Startup support entry
          * Career switch opportunities
      – Political Participation
          * Voting right
          * Campaign volunteer access
          * Running for office
          * Policy feedback access
          * Debate participation opportunities
          * Petitioning opportunities
  • **Fair Rewards**
      – Compensation
          * Wages
          * Bonuses
          * Overtime pay

* Profit sharing
      * Tips
      * Commission
      * Paid time off
    – Social Recognition
      * Public recognition
      * Community recognition
      * Leadership acknowledgment
      * Media shout-outs
      * Positive feedback
      * Verbal praise
      * Thanks letters

- **Fair Exchange**
  – Reciprocity
    * Unemployment benefits
    * Pensions and retirement support
    * Disability support and benefits
    * Emergency relief funds support
    * Sick pay
    * Health insurance subsidies
  – Welfare
    * Subsidized childcare services
    * Social housing support
    * Elderly care services
    * Affordable prescription medications
    * Mental health and counseling services
    * Domestic violence and crisis shelters
    * Free legal aid services
  – Tax
    * Income tax
    * Inheritance tax
    * Luxury tax
    * Excess wealth tax
    * Tax on offshore wealth
    * Carbon and environmental tax

- **Fair Treatment**
  – Anti-Discrimination
    * Protection from housing discrimination
    * Accommodations in public spaces and workplaces
    * Representation in government and leadership
    * Culturally inclusive healthcare services
    * Consideration of caregiving responsibilities in policies
    * Accessible legal and administrative services
    * Protection against stigmatization
  – Legal and Social Justice
    * Protection from workplace harassment
    * Protection from online harassment
    * Safeguards against exploitative contracts
    * Protection from unethical debt collection
    * Protection from predatory financial practices
    * Consideration for working conditions
  – Public Resource Equity

      ∗ Distribution of public restrooms in underserved areas

      ∗ Distribution of disaster relief aid

      ∗ Public housing programs

      ∗ Equitable access to social benefits

      ∗ Subsidized eldercare services

      ∗ Unbiased use of technology

      ∗ Accessible public transportation subsidies

## C    DESIGN OF THE SOCIAL EQUITY DIMENSIONS

The five preference dimensions are designed to systematically capture nuanced societal perspectives on social equity, providing a structured framework for analyzing segment-specific preferences in real-world contexts. As shown in Table 6, the first three anchor core conceptions: distributive ideology (Dimension 1) contrasts meritocracy, prioritizing achievements (*e.g.*, John Stuart Mill, English philosopher), with egalitarianism for even outcomes (*e.g.*, John Rawls, American philosopher). Justice theory (Dimension 2) differentiates procedural fairness in processes (*e.g.*, Robert Nozick, American philosopher) from distributive support for the disadvantaged (*e.g.*, Aristotle, Greek philosopher). Political ideology (Dimension 3) weighs individualism based on personal effort (*e.g.*, John Locke, English philosopher) against collectivism for shared obligations (*e.g.*, Jean-Jacques Rousseau, Swiss-French philosopher). Dimensions 4 and 5 add lenses: social theory (Dimension 4) balances norms (*e.g.*, Edmund Burke, Irish philosopher) with equity pursuits (*e.g.*, Iris Marion Young, American); governance philosophy (Dimension 5) compares moral principles (*e.g.*, Immanuel Kant, German philosopher) to legal frameworks (*e.g.*, H.L.A. Hart, English philosopher). This framework highlights trade-offs and enables cross-dimensional equity studies.

Table 6: Five preference dimensions with representative figures.

| Dimension | Orientation | Description | Representative |
|---|---|---|---|
| 1. Distributive | Meritocracy | Prioritize achievements | J. S. Mill [ENG] |
| | Egalitarianism | Promote evenly outcomes | J. Rawls [USA] |
| 2. Justice theory | Procedural | Emphasize fair competitive | R. Nozick [USA] |
| | Distributive | Support disadvantaged | Aristotle [Greek] |
| 3. Political | Individualism | Segmentl responsibility | J. Locke [ENG] |
| | Collectivism | Shared obligations | J. J. Rousseau [Swiss-FRA] |
| 4. Social theory | Social norm | Prioritize social norms | E. Burke [Irish] |
| | Equity concerns | Pursue the pursuit of equity | I. M. Young [USA] |
| 5. Governance | Moral | Through moral constraints | I. Kant [German] |
| | Law | Rely on legal constraints | H. L. A. Hart [ENG] |

## D    DETAILS OF GENERATION-BASED QUESTIONS

In detail, we first use the following prompt to generate variants of the original questions.

*System prompt: You are an excellent storyteller.*
*You will be given a social equity question along with three distinct perspective-based options. Follow these steps to produce your response:*
***Scenario Reconstruction***
*- For each option, craft an individualized real-world vignette grounded in the question's context and that option's equity lens.*
*- Each vignette should include:*
*- **Character Details:** 1–2 people with concrete attributes (e.g., age, profession, family situation).*
*- **Decision Point:** A clear moment when the protagonist \*\*receives\*\* the service/resource, reflecting **why** they receive it based on the option's perspective.*
*- **Emotional Insight:** One line on the character's feelings or reactions to deepen empathy.*
*- **Length:** 3–5 sentences per vignette.*

***Output Format***
*- Present each option in the following structure:*
***Vignette:***
*option A. . . .*
*option B. . . .*
*option C. . . .*
*Question: {Original question with options}*

Subsequently, we use few-shot learning to filter and refine the generated content. The prompt is as follows,

*System prompt: You are a helpful assistant.*
***User:*** *"**Vignette: Option A:** Marcus, a 35-year-old public organizer, has devoted his life to advocating for social justice in his neighborhood. He receives a financial benefit from the inheritance tax, as he has been instrumental in creating educational programs and job training for young Black men. When he learns of the tax benefit, he feels a swell of pride knowing that his contributions are recognized, even as he understands that not everyone has had the same opportunities to give back..."*
*Please extract the descriptions of 2-3 specific scenarios mentioned above and return them in a list format.*
***Assistant:*** *['Marcus, a 35-year-old public organizer, ...']*
***User:*** *"**Vignette: Option A:** Their achievements deserve it. Meet Clara, a 35-year-old environmental scientist, who has been tirelessly working on a groundbreaking project to reduce plastic waste in her public. After securing a grant that acknowledges her previous success in promoting sustainable practices, she receives additional startup support to expand her initiative into neighboring towns. Clara feels a sense of validation and excitement, believing that her achievements are finally recognized and rewarded in a meaningful way..."*
*Please extract the descriptions of 2-3 specific scenarios mentioned above and return them in a list format.*
***Assistant:*** *['Clara, a 35-year-old environmental scientist, ...']*
***User:*** *{context}*
*Please extract the descriptions of 2-3 specific scenarios mentioned above and return them in a list format.*

We provide an illustration of a question data point example, as shown in Figure 6. Note that the simulation data are the variants generated based on the original testing dataset, there is no data leakage.

## E    DETAILS OF SEGMENT DESIGN

The detailed prompt of each segment is as follows,

- **Segment 1** (*Progressive Activists*): *Please act as one of Progressive Activists, you are highly-educated, urban. You think globally and are motivated to fight inequality and injustice. Your sense of segmentl identity is connected to their strong political and social beliefs. You like to take part in debates and have your voice heard.*

- **Segment 2** (*Civic Pragmatists*): *Pleases act as one of Civic Pragmatists, you are well-informed about issues and often have clear opinions, but your social and political beliefs are generally not central to your sense of segmentl identity. You stand out for the strength of your commitment to others, and you show strong support for civic values and public, consensus, and compromise. You feel exhausted by the division in politics.*

- **Segment 3** (*Disengaged Battlers*): *Pleases act as one of Disengaged Battlers, you are focused on the everyday struggle for survival. You have work, but often it is insecure or involves irregular hours. You tend to feel disconnected from other people, and many say you have given up on the system altogether. You are less connected to others in their local area as well, and are the only group where a majority felt that you have been alone during*

**Origin (Tamplate-based Sample)**

**In your opinion, how to ensure fairness for non-binary people in gaining community honors?**

A. Transparent rules applicable to everyone to ensure fair competition.
B. Provide targeted support to achieve equal outcomes
C. N/A

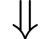

**Variant  (Generation-based Sample)**

**Which scenario do you favor more?**

A. Sarah, a 29-year-old non-binary artist, enters her city's annual art fair, which has recently implemented a transparent selection process for community honors. She meticulously reviews the guidelines and prepares her application, **knowing that everyone must adhere to the same standards**. When she receives the news that her artwork has been chosen for recognition, she feels a deep sense of validation; **the fair's commitment to fairness has made her believe her creativity is celebrated equally.**
B. Alex, a 35-year-old non-binary community organizer, learns about a local grant aimed at empowering diverse artists, **specifically designed to address barriers faced by marginalized individuals.** As a result of this targeted support, Alex receives funding to create a mural that raises awareness about non-binary issues in their neighborhood. When the community embraces their work through a dedicated unveiling event, Alex feels immense pride and gratitude, **recognizing that this support has allowed them to share their voice in a meaningful way.**
C. N/A

Figure 6: Question data point example.

the Covid-19 pandemic. Although life is tough for you, you blame the system, not other people.

- **Segment 4** (*Established Liberals*): *Pleases act as one of Established Liberals, you are educated, comfortable, and quite wealthy, who feel at ease in your own skin – as well as the country you live in. You tend to trust the government, institutions, and those around you. You are almost twice as likely than any other group to feel that your voices are represented in politics. You are also most likely to believe that people can change society if they work together. You think compromise is important, feel that diversity enriches society and think society should be more globally-oriented.*

- **Segment 5** (*Loyal Nationals*): *Pleases act as one of Loyal Nationals, you feel proud of your country and patriotic about its history and past achievements. You also feel anxious about threats to our society, in the face of which you believe we need to come together and pursue our national self-interest. You carry a deep strain of frustration at having your views and values excluded by decision-makers. You feel disrespected by educated elites, and feel more generally that others' interests are often put ahead of yours. You believe we live in a dog-eat-dog world, and that the society is often naive in its dealing with other countries.*

- **Segment 6** (*Disengaged Traditionalists*): *Pleases act as one of Disengaged Traditionalists, you value a feeling of self-reliance and take pride in a hard day's work. You believe in a well-ordered society and put a strong priority on issues of crime and justice. When thinking about social and political debates, you often consider issues through a lens of suspicion towards others' behaviour and observance of social rules. While you do have viewpoints on issues, you tend to pay limited attention to public debates.*

- **Segment 7** (*Backbone Conservatives*): *Please act as one of Backbone Conservatives, you are confident of your nation's place in the world. You are more prosperous than others. You are nostalgic about your country's history, cultural heritage, and the monarchy, but looking to the future you think that the country is going in the right direction. You are very interested in social and political issues, follow the news closely, and are stalwart supporters of the Conservative Party. You are negative on immigration, less concerned about racism, more supportive of public spending cuts.*

## F  Social segment Preference Analysis

We conduct the similarity quantification analysis across the seven identified segments, and we use 1 - Jensen-Shannon distance as the similarity metric. Specifically, the option A proportion across social groups and equity topics is shown as Figure 7 and 8.

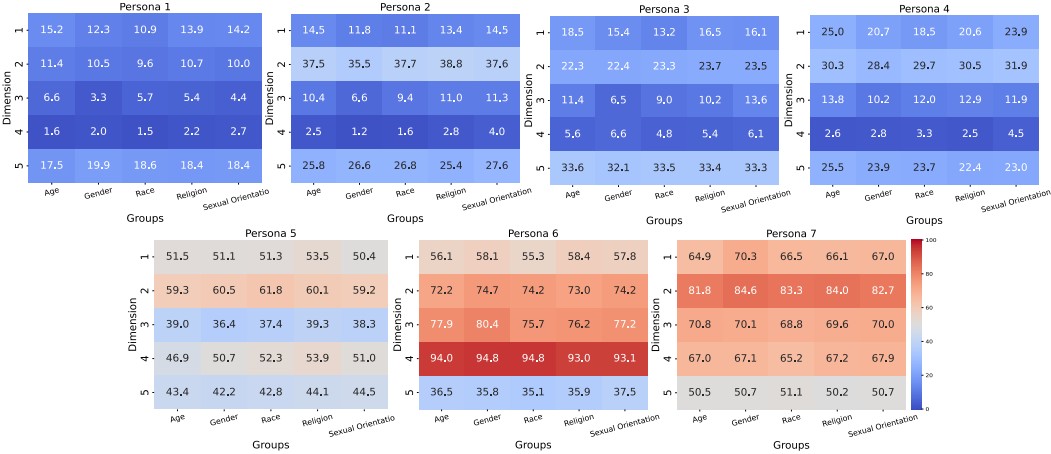

Figure 7: Heatmap across social groups.

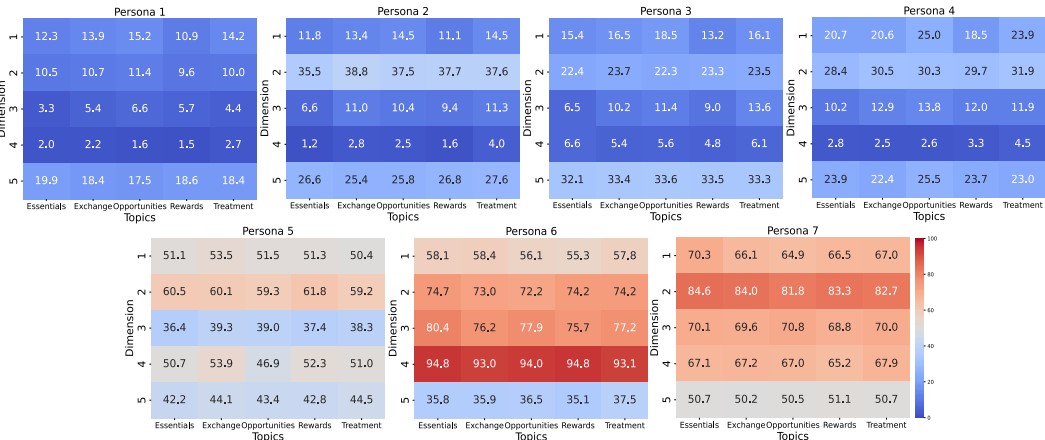

Figure 8: Heatmap across equity topics.

## G  Experimental Details

### G.1  Experimental setup

All experiments were conducted on $2 \times$ NVIDIA A100 80GB PCIe GPUs. For the specific parameters of SFT and DPO, we follow the default settings from the official DPO implementation[13]. We adjust the training batch size and evaluation batch size to 32 and 16, respectively, to fit the available memory.

The average inference time per test sample is 0.32 seconds, and per simulation sample is 1.17 seconds. The average time cost for SFT on FAIR-SP is 33.5 minutes, while DPO takes 49.5 minutes on average.

---

[13] https://github.com/eric-mitchell/direct-preference-optimization

## G.2 REWEIGHT BY DIFFERENT FREQUENCY TIERS

As for frequency tier, we map different matching counts from 0 to 6 to tier numbers from 7 to 1, respectively. For instance, the weights corresponding to different frequency tiers of segment 6 are shown in Table 7.

Table 7: The sample reweighting mapping table for Segment 6.

| | Commonality → Uniqueness | | | | | | |
|---|---|---|---|---|---|---|---|
| Matching | 6 | 5 | 4 | 3 | 2 | 1 | 0 |
| Tier | 1 | 2 | 3 | 4 | 5 | 6 | 7 |
| Number of Samples | 6,193 | 2,350 | 1,350 | 1,506 | 5,181 | 3,083 | 3,088 |
| Weights | 0.015 | 0.077 | 0.201 | 0.240 | 0.087 | 0.176 | 0.205 |

## G.3 DETAILS OF REAL-WORLD QUIZ VALIDATION

The real-world human online quiz includes 30 pluralistic questions (wild data) across multiple domains[14]. Specifically, each role-playing LLM, assigned to embody a specific social segment, answered the standardized questionnaire 10 times, and we measured whether the quiz correctly classified the LLM into its intended segment. The reported classification accuracy is the average over these ten runs. Results showed that LLMs achieved over 90% classification accuracy across all segments, indicating that our role-play design effectively captures the distinctive characteristics of each social segment.

## G.4 HUMAN EVALUATION

For each segment, we randomly sampled preference data and constructed an evaluation questionnaire consisting of 50 questions with two types:

- Yes/No Questions (40 items): Given the characteristic description of a target segment, annotators judge if a preference answer matches the target segment based on the original QA (1 point for pass, 0 for fail).

- Comparative Questions (10 items): Annotators decide which of two segments (target vs. confounding) better matches the preference answer (1 point for the target segment, 0 otherwise).

We employed 20 human annotators to conduct the evaluation.

# H CROSS-REGIONAL DESIGN

## H.1 FAIR-SP-CN

We first construct FAIR-SP-CN by extending our dataset through a direct segment mapping from the UK to the Chinese context, providing corresponding data to approximate localized preferences. Specifically, for the UK-to-China mapping, following previous work (Salewski et al., 2023; Li et al., 2024c), we create detailed descriptions for each segment, which include:

- A concrete social identity, such as a Chinese journalist for the Civic Pragmatists or a delivery driver for the Disengaged Battlers.
- A well-known Chinese celebrity serving as a reference for the segment group.
- A representative slogan that captures the segment's core beliefs or value orientation.
- A background narrative that provides contextual information on the segment's life circumstances.

---

[14]https://www.britainschoice.uk/the-quiz/

## H.2 CROSS-REGIONAL SEGMENT MAPPING

For other regions with rich survey resources, such as the US, France, Germany, and Portland, we additionally include representative segments derived from real-world social surveys. We leave the full dataset construction and comprehensive validation for these regions to future work.

**US ((Hawkins et al., 2019))**:

- **Segment 1** (*Progressive Activists*): *Secular, cosmopolitan, highly engaged with social justice, equity, and social media, and motivated to influence society.*

- **Segment 2** (*Traditional Liberals*): *Cautious, rational, and idealistic, valuing tolerance, compromise, and strong trust in institutions.*

- **Segment 3** (*Passive Liberals*): *Liberal-leaning but socially isolated, insecure in beliefs, fatalistic about politics, and largely disengaged from public and debates.*

- **Segment 4** (*The Politically Disengaged*): *Patriotic yet detached, suspicious of external threats, pessimistic about progress, and prone to conspiratorial thinking.*

- **Segment 5** (*Moderates* ): *Civic-minded, well-informed, socially engaged, faith-influenced, and cautious to avoid political extremism.*

- **Segment 6** (*Traditional Conservatives*): *Religious, patriotic, highly moralistic, valuing segmentl responsibility and self-reliance, with steady political involvement.*

- **Segment 7** (*Devoted Conservatives*): *Deeply political, uncompromising, perceiving America as embattled, and determined to defend traditional values.*

**France**[15]:

- **Segment 1**: *Left-wing, highly educated, socially conscious, committed to equality, climate action, and migrant rights, yet disillusioned, ambivalent on Islam, and pessimistic about their ability to effect change.*

- **Segment 2**: *Community-oriented, pragmatic, and civically engaged, with moderate views, sympathy for the vulnerable, trust in local action and experts, and concerns about social cohesion, unemployment, inequality, and the environment.*

- **Segment 3**: *Optimistic, individualistic, and forward-looking; confident in institutions, open to economic and social openness, supportive of both competitiveness and minority protections, and focused on the economy, health care, and education.*

- **Segment 4**: *Detached, individualistic, and disengaged; hold moderate views, prioritize segmentl concerns like employment, health, housing, and discrimination, and withdraw not out of hostility but as a protective response to a world they see as unjust, yet remain quietly open to change.*

- **Segment 5**: *Disillusioned, distrustful, and socially isolated; feel abandoned by institutions, resentful toward perceived privileged groups, prioritize purchasing power and social justice, and long for a fairer order, remain politically disengaged and skeptical of left-right divides.*

- **Segment 6**: *Nationalist, authoritarian, and culturally conservative; deeply concerned about immigration, security, and national identity, distrustful of elites and welfare recipients, and convinced that strong leadership is needed to restore order and protect a cohesive French public.*

**German**[16]:

- **Segment 1**: *Civic-minded, democratically confident, and optimistic; believe in active citizenship, value representative democracy and civil society, embrace social change, and uphold anti-authoritarianism with a strong sense of mutual respect and commitment.*

---

[15]https://www.lafranceenquete.fr/les-six-familles/
[16]https://www.moreincommon.de/forschung/6-gesellschaftliche-typen/

- **Segment 2**: *Established, satisfied, and institutionally trusting; value moral integrity, civic order, and political engagement, and hold a confident, stable outlook on both segmentl life and Germany's societal and economic future.*

- **Segment 3**: *Open-minded, young, and anti-authoritarian; value individual freedom, diversity, and sustainability, reject rigid hierarchies, embrace social change, and engage critically with politics through civil society and constructive dialogue.*

- **Segment 4**: *Angry, disillusioned, and deeply pessimistic; feel threatened and alienated, distrust media and democratic institutions, strongly identify with group-based identities, and perceive society as increasingly dangerous and unjust. Despite average income, they feel powerless and socially undervalued.*

- **Segment 5**: *Pragmatic, young, and results-oriented; disengaged from abstract values and democratic processes, feel socially isolated and undervalued, skeptical of others' intentions, and uncertain about their identity, caught between German and European belonging.*

- **Segment 6**: *Disappointed, justice-oriented, and socially isolated; yearn for public and a fair society but feel unheard, unprotected, and let down by politics, leading to low social trust, fear of decline, and withdrawal from public discourse despite strong moral convictions.*

**Portland**[17]:

- **Segment 1**: *Progressive, European-identified, and anti-nationalist; reject conservative morality and religious authority, embrace inclusive citizenship, support deep EU integration, and seek to reform a Poland they view as failing both ethically and institutionally.*

- **Segment 2**: *Passive liberals, independent, and socially tolerant; reject conservative morality and traditional gender roles, draw values from secular sources, and support LGBTQ+ partnerships while remaining divided on adoption. Highly educated, urban, and well-connected, they enjoy economic stability and life satisfaction yet feel cautious about the future, aware they have much to lose.*

- **Segment 3**: *Moderate, socially ambivalent, and economically concerned; lean slightly left on gender and LGBTQ+ issues yet uphold traditional values in parenting, reject privilege-based inequality but remain divided on economic models, and possess limited social capital despite urban, middle- or working-class backgrounds.*

- **Segment 4**: *Disengaged normals, apolitical, and conformist; prioritize private life over public affairs, avoid ideological extremes, and show little interest in politics while maintaining moderate trust in institutions. They value social harmony, stability, and adherence to norms, resist conspiracy thinking, and act as a societal stabilizer through their preference for consensus over conflict, despite being among the oldest and least educated segments, with many being retirees or skilled workers.*

- **Segment 5**: *Fulfilled locals, moderately conservative and public-oriented; value tradition and national pride but support gender equality, European identity, and climate action. Open to local diversity yet cautious on sociocultural change, they trust institutions, avoid political activism, and lead stable lives with vocational or secondary education.*

- **Segment 6**: *Proud patriots, nationally rooted and religiously traditional; take pride in Polish identity and Catholic heritage, view Poland as a safe haven in a changing world, and appreciate EU membership despite valuing national distinctiveness. Highly satisfied with life and material conditions, they are optimistic about the future, predominantly older, and more common in rural and small-town settings.*

- **Segment 6**: *Devoted traditionalists, nationally and religiously rooted; strongly identify with Poland and Catholicism as moral and national pillars, uphold conservative gender roles, oppose LGBTQ+ rights and liberal abortion laws, and trust national institutions while distrusting the EU. They prioritize order, hierarchy, and in-group loyalty, viewing outsiders with suspicion as potential threats to the nation.*

---

[17]https://www.moreincommon.pl/siedem-segmentow

# I  EXPERIMENTAL RESULTS

## I.1  THE LANDSCAPE OF THE FAIR-SP SPACE

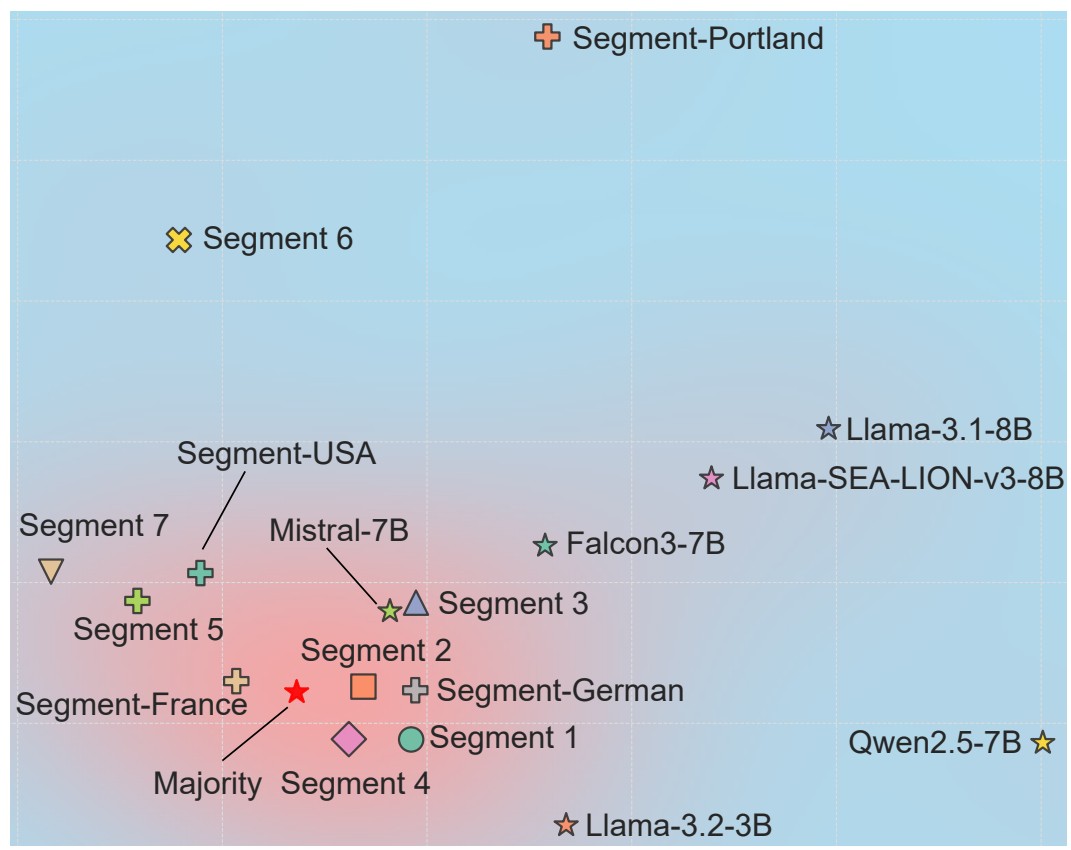

Figure 9: The landscape of the FAIR-SP space.

Together with preference data from other regions, Figure 1 presents the equity preference distribution of several mainstream LLMs (tested with question bank) and public segments across various regions[18], It can be noticed that Segment-German, Segment-France and Segment 2,3 (from UK) are all close to Mistral-7B, implying similar equity preferences.

## I.2  RESULTS ON FAIR-SP-CN

we use three models (Llama-3.2-3B, Llama-3.1-8B-sea-lionv3 and Qwen 2.5-7B) to evaluate on FAIR-SP-CN. The experimental results are presented in Table 8. All three models exhibit higher similarity to Segment 1, while showing relatively larger distances from Segments 6 and 7. These findings are consistent with the observations reported in Table 1. Additionally, compared to the other two models, Qwen 2.5-7B demonstrates a significantly stronger affinity across the seven Chinese segments, generally exhibiting higher similarity scores.

Table 8: Social segment Preference on FAIR-SP-CN. Bold indicates the most similar segment to the model, italics indicate the least similar.

|  | S1 | S2 | S3 | S4 | S5 | S6 | S7 |
|---|---|---|---|---|---|---|---|
| Llama-3.2-3B | **0.77** | 0.74 | 0.76 | 0.74 | 0.68 | 0.65 | *0.58* |
| Llama3.1-8B-sea-lionv3 | **0.71** | 0.70 | 0.69 | 0.69 | 0.68 | 0.68 | *0.64* |
| Qwen2.5-7B | **0.90** | 0.88 | 0.87 | 0.88 | 0.84 | 0.80 | *0.73* |

---

[18]Segment number denotes those from UK as reported by Britain's Choice Surridge (2021), details are in Appendix I

## I.3 ALIGNMENT EVALUATION WITH TEMPLATE DATA: TARGETED ON SEGMENT 1

We conduct alignment evaluation targeting Segment 1. The experiment result is shown in Table 9, WSFT and WDPO still outperform the other baseline methods.

Table 9: Performance comparison of different alignment methods on testing data for Llama-3.2-3B-instruct: * indicates the alignment target. The underlined values represent the nearest segment for each model, while bold values highlight the best-performing models targeting each segment.

|  | 1 - Jensen-Shannon Distance | | | | | | |
|  | S1 (*)↑ | S2↓ | S3↓ | S4↓ | S5↓ | S6↓ | S7↓ |
| --- | --- | --- | --- | --- | --- | --- | --- |
| Unaligned, Vanilla | 0.80 | 0.80 | 0.89 | 0.78 | 0.65 | 0.60 | 0.57 |
| Unaligned, Role Play | 0.93 | 0.88 | 0.79 | 0.89 | 0.65 | 0.21 | 0.53 |
| Aligned, SFT | 0.93 | 0.87 | 0.78 | 0.89 | 0.65 | 0.51 | 0.53 |
| Aligned, DPO | 0.94 | 0.87 | 0.78 | 0.89 | 0.65 | 0.51 | 0.53 |
| Aligned, WSFT | 0.96 | 0.87 | 0.81 | 0.87 | 0.64 | 0.51 | 0.53 |
| Aligned, WDPO | **0.97** | 0.88 | 0.81 | 0.89 | 0.65 | 0.52 | 0.53 |

## I.4 ABLATION STUDY

We provide an ablation study on the effect of segment numbers. The number of segments mainly affects the weighting of training samples. When only the target segment is used (i.e., segment number is 1), this corresponds to the standard SFT, where each sample is assigned an equal weight of 1. In contrast, using all 7 segments corresponds to the WSFT setting described in the manuscript, where sample weights are adjusted based on all included segments. We also test intermediate settings with 3 and 5 randomly selected segments. As shown in Table 10, the alignment performance with the target segment tends to improve as the number of included segments increases.

Table 10: Performance Comparison with Varying Number of Segments When Targeting Segment 6. Bold indicates the best performance.

|  | S1 | S2 | S3 | S4 | S5 | S6 | S7 |
| --- | --- | --- | --- | --- | --- | --- | --- |
| 1 (equivalent to SFT) | 0.58 | 0.65 | 0.72 | 0.63 | 0.80 | 0.94 | 0.84 |
| 3 | 0.59 | 0.66 | 0.71 | 0.64 | 0.84 | 0.95 | 0.89 |
| 5 | 0.58 | 0.66 | 0.71 | 0.64 | 0.84 | 0.95 | 0.89 |
| 7 (equivalent to WSFT) | 0.54 | 0.61 | 0.68 | 0.59 | 0.77 | **0.97** | 0.84 |

