# OpenReview forum: "Fair-SP: Capturing Pluralistic Social Equity Preferences Through Synthetic Data"
_ICLR.cc/2026/Conference — ICLR 2026 Conference Withdrawn Submission_

### Official Review · Reviewer_sYz9 · 2025-10-14

**Soundness:** 3
**Presentation:** 3
**Contribution:** 2
**Rating:** 4
**Confidence:** 4

**Summary:**

The authors contribute a large synthetic dataset for pluralistic alignment on social equity issues, where GPT-4o role plays as one of seven representatives "types" from the UK. They validate that GPT-4o successfully roleplays as a stereotype of the group, and also do an experiment training models to steer towards the data.

**Strengths:**

S1: The dataset is very large, and focuses specifically on equity, a focus I've not seen before in a pluralistic alignment dataset.

S2: The experimental methodology is sound, and I feel that the authors did a good job attacking theh problem from various angles.

**Weaknesses:**

W1: The paper discusses the preferences as a contribution - however, all preferences are generated by GPT-4o-mini, and not by actual people. Additionally, there have been several works (e.g, Randomnness, Not Representation https://arxiv.org/abs/2503.08688, The Illusion of Artificial Inclusion https://dl.acm.org/doi/pdf/10.1145/3613904.3642703) questioning the validity of substituting human response data with machine generated data. While I do applaud the paper's "Role-play fidelity validation," I think that while it does show that, in broad strokes, GPT4o-mini can simulate a stereotypical member of the group, that is not quite the same as actually capturing the preference variation from the group. Additionally, the More in Common quiz and the preference questions are different distributions, limiting the fidelity of the preference data in my opinion. The human validation as well was from external annotators, not from people of the actual group, limiting the conclusions we can draw from the evaluation. In my opinion, these synthetic preferences would be very strengthened by validating it against actual human opinion data.

W2: Limited applicability to LLM tasks. All of the experiments focus on multiple choice survey data. However, many LLM tasks are not multiple choice, and LLMs are not as consistent as humans, weakening the general applicability of multiple choice data. This is not fleshed out as a limitation as currently stated, and should be engaged with.

W3: Limited motivation. While there has been a recent focus on pluralism in alignment,  the authors do not engage very much with what goals they are trying to achieve in this work, beyond constructing a synthetic dataset. What goals do you hope that the dataset achieves? How do you hope that the dataset will be used? Because of W1/W2, I'm afraid that the fact that the dataset is synthetic and limited to a relatively narrow sub-domain limits the applicabiliity of the work.

W4: While the questions seem to vary quite a bit, the answers seem to be drawn from a very small set, limiting the diversity of the dataset.


There are some minor clarity issues:
- Minor point - I was slightly confused by the "1 - JS Distance" terminology at first. I wonder if it might be more clear to define it as the "Jensen Similarity" and refer to it as such. Or, alternatviely, you could just report the JS Distance and flip the desired directionality (minimization)
- "human evaluation scores exceeding 0.7" in the Abstract and one other place is confusing - what is the unit?

Additionally, there are some problems with the relationship with related work:
- MaxMinRLHF (https://arxiv.org/pdf/2402.08925) is highly relevant, dealing with subgroup preferences on multiple choice questions. It would be good to engage with similarities and differences
- L449 states that Feng et al introduce the ValuePrism dataset. While they do utilize the ValuePrism dataset, it is actually introduced in another paper: Value Kaleidoscope (https://arxiv.org/abs/2309.00779)

**Questions:**

Q1: Can you say more about the intended impact of the work? Is it meant to be a diagnostic for LMs, or a measure of their steerability to a particular group, or a resource for training, or something else?

---

### Official Review · Reviewer_5wTU · 2025-10-29

**Soundness:** 1
**Presentation:** 2
**Contribution:** 1
**Rating:** 2
**Confidence:** 4

**Summary:**

The paper presents FAIR-SP, a new synthetic dataset of preferences about situations involving equity concerns. The dataset simulates the preferences of 7 UK "social segments" on 98 equity related issues regarding 28 "social groups" (e.g. race, ethnicity but also groups like 9/11 victims) across 5 preference dimensions via gpt-4o-mini.

**Strengths:**

The paper introduces FAIR-SP, a synthetic dataset designed to broaden the range of preferences typically represented in such datasets and enhance pluralistic perspectives. A strength of the paper is that it attempts to ground choices about the dataset in existing social science work, however, as described in the weaknesses, the engagement with the social sciences remains superficial.

**Weaknesses:**

# No validation of LLM simulations with people in the simulated social groups

The biggest problem that I see in this paper is the use of an LLM (gpt-4o-mini) to simulate the preferences of many varied social segments, for high-stakes scenarios related to equity, but without any validation with people who are actually a part of those social segments. The human evaluation that the paper includes is an evaluation with 20 annotators---who are not necessarily a part of those social groups---and asks them whether the preference is reflective of the segment. So, in other words, the LLM simulates the preferences of a social group, and the humans also simulate what they think this social group thinks. This approach introduces significant potential for bias and stereotyping.

The authors simply state that "Given that advanced LLMs demonstrate strong role-play capabilities, we leverage GPT-4o-mini" and do not engage with any of the literature showing that LLM simulation of human preferences leads to stereotyping and inaccuracies. It is important for the authors to address this body of work and provide empirical validation for their simulations to counter these concerns. Here is one reference that the users can use as a starting point:

> "Large language models that replace human participants can harmfully misportray and flatten identity groups" Wang et al. Nature Machine Intelligence 2025.

While the authors show that gpt-4o-mini, when given a quiz that classifies humans into these segments, is accurate at receiving the correct segment classification, this doesn' tmean it is accurate at simulating all the questions that the authors use in teh dataset.

# Superficial connection to social sciences

While the authors claim that they ground their dataset in the social sciences, their engagement with the social sciences also seems superficial. As an example, in describing the five preference dimensions they consider, the authors say they "draw on sociology, political theory, ethics, and cultural psychology". When reading this, I expected them to draw on some systematic and cohesive framework, e.g., Hofstede's cultural dimensions, moral foundations theory, Schwartz's theory of basic human values, etc. However, it appears that the authors selected each of the five dimensions independently, based on their own reading of various literatures, without providing further detail on how these specific dimensions were chosen.

# Existing work
It also doesn't make sense to me that the paper focuses on simulating _synthetic_ preferences for seven broad UK segments when the PRISM dataset (which the authors cite) already includes preferences from demographically-representative human annotators in the UK. While the prompts may differ between PRISM and the FAIR-SP dataset, PRISM seems like it has more ecological validity because the human annotators were asked to write their own prompts, focusing on something related to their values or a controversial issue. This approach seems more authentic and reflective of real-world LLM use cases.

**Questions:**

How did the authors choose the 5 dimensions they focused on? How did they come up with the minority social groups (e.g. "9/11 victim")?

---

### Official Review · Reviewer_wnRk · 2025-11-03

**Soundness:** 2
**Presentation:** 2
**Contribution:** 2
**Rating:** 2
**Confidence:** 4

**Summary:**

The authors introduce FAIR-SP, a synthetic dataset for studying and aligning LLMs to pluralistic social-equity preferences. FAIR-SP spans a range of social groups, equity topics, and preference dimensions by generating 34K multiple-choice questions and 238K preference decisions labeled by GPT-4o-mini conditioned on seven UK public segments. The authors validate the synthetic generations with alignment on the “Britain’s Choice” segmentation quiz and run small-scale human evaluations (20 annotators) and find 70% annotator agreement with the generated preferences. They then demonstrate dataset utility by aligning models toward a target segment using SFT/DPO and a reweighting variant that emphasizes segment-unique samples, reporting higher similarity to the target and increased margins from other segments. Overall, the paper aims to provide a reusable, simple technique of synthetic data generation for pluralistic alignment.

**Strengths:**

S1. There is clear motivation for this work. The paper articulates why existing preference datasets under-represent pluralism in the UK context across public segments and multiple value dimensions.

S2. Human validation. The addition of the segmentation quiz to show alignment between synthetic preferences and more importantly human evaluation with 70% agreement rates provides some preliminary evidence that segment portrayals and question vignettes are recognizable. Having human evaluation that is representative of the segments being analyzed is important in alignment research given the brand new nature of many of these product types and the unknowns around how strongly existing survey preferences correlate with preferences when it comes to chatbot conversations.

S3. The demonstrations of the practical utility of the dataset are welcome additions. The fine-tuning experiments with DPO/SFT and the weighted versions which show the ability to use the

**Weaknesses:**

W1. The group/segment ground truth and portability are under-substantiated. Seven UK-centric segments are treated as universal anchors without much validation. Mapping them cross-regionally (e.g. to China in the appendix) is interesting but lacks prospective human survey evidence showing that the same axes/questions cleanly transfer to other groups or that the idea of basing anchors solely on a set of arbitrarily chosen human value surveys is a reasonable approach for achieving high coverage of all possible human values during preference dataset collection.

W2. The authors develop synthetic preferences in sensitive demains which could introduce circularity, primarily encoding ideas based on survey aggregations which could further exacerbate bias. It’s not clear if the current human agreement rates (70%) are good or bad or even if those humans were representative of the population.

W3. In terms of population representativeness I have some serious concerns given that the study is focused on the UK but the paper is showing results that are clearly not census-representative, for example including Latino, American Indian, and African American in the race demographic in Figure 4. It’s also unclear why inSsection 3 the authors then define representativeness in a global fashion, selecting models as "representative" based on where the team that trained them is based (e.g. Mistral vs Falcon) rather than based on their training corpus or any targeted post-training.

W4. Limited auditing for stereotyping and normativity. The taxonomy includes sensitive groups (e.g. religions, sexual orientations). The paper does not show comprehensive bias audits, harm analysis, or red-teaming of the narrative generator/role-play outputs to ensure scenarios don’t encode or reinforce stereotypes.

W5. Alignment results risk overfitting to anchors. WDPO/WSFT gains are reported largely in terms of greater similarity to the target segment and separation from others. Without out-of-segment human evaluation (e.g. raters judging whether outputs are aligned with a given segment on new equity prompts) or downstream task impacts, it’s hard to know whether we’re aligning meaningfully or just optimizing a proxy.

W6. The related-work positioning could be tighter: the paper cites PRISM/CulturePark/etc and pluralistic alignment methods like GRPO, but the novelty delta relative to these is not quantified through metrics like overlap of topic spaces, annotation sources, or alignment outcomes under a shared metric/benchmark.

**Questions:**

Q1. Will the dataset, prompts, and code be publicly released and was any IRB/ethics review undertaken given the sensitive framing around protected groups? Any additional plans for safeguards around the data and release process would be welcome additions.

Q2. How stable are segment classifications under prompt or temperature variations? Reporting variance across at least 5 random seeds or temperature sampling for the role-play fidelity and downstream JS metrics would help improve validation.

Q3. The baseline segmentation is strongly grounded in a specific More in Common survey and the five fairness dimensions from The Fairness Foundation. It would be helpful to understand how these surveys were conducted and how they compare with other commonly cited works like the World Values Survey in terms of their size, survey representativeness, and focus.

---

> ### Comment · Reviewer_wnRk · 2025-11-13
>
> I've read through the other reviewers feedback and there seems to be clear consensus that this work as-is should be rejected. If the authors have follow-ups on the questions or weaknesses identified that would help them in iterating for future submissions I'm happy to continue to answer further questions.

---

### Official Review · Reviewer_rvhk · 2025-11-05

**Soundness:** 1
**Presentation:** 1
**Contribution:** 2
**Rating:** 2
**Confidence:** 4

**Summary:**

This paper introduces Fair-SP, a dataset of questions templated as “How to ensure fairness for {group} to receive the benefits of {topic}”, covering 98 equity-related topics, 28 social groups, and 10 value dimensions grounded in established theoretical frameworks across multiple disciplines. A subset of the dataset is further extended into narrative-driven, scenario-based questions that feature characters and real-world settings to enhance contextual nuance. The paper shows that this dataset can be used to examine how different population segments express equity-related stances through prompt-based LLM role-playing. They also explore which social segments various LLMs are most aligned with and analyze how preference learning can steer (or fail to steer) LLMs toward specific population preferences.

**Strengths:**

- The work has strong potential to become a valuable resource for studying pluralistic social values and equity in LLMs.
- The authors make a commendable effort to comprehensively cover a wide range of equity topics, value dimensions, social groups, and population segments, informed by sociological and philosophical theories.
- The paper goes beyond simple template-based question generation by incorporating scenario-based, narrative questions that better reflect how fairness issues manifest in real-world contexts. This addition substantially enhances the dataset’s depth and realism.
- The authors conduct extensive experiments to validate multiple aspects of the dataset and its potential applications, including equity distribution analyses, segment-based alignment, and human evaluation.

**Weaknesses:**

- Lack of motivation for the dataset and task: The introduction does not convincingly establish why this dataset is needed or what specific research gap it fills. The paper should more clearly justify the importance of creating “narrative-driven contextualized scenarios” and explain why this feature matters for studying fairness or alignment.
- Weak motivation for methods and analyses: Several methodological sections (e.g., 3.2.1, 3.1, 3.2.3) lack clear justification for the design choices. As a result, the paper feels fragmented, and the overall contribution does not form a coherent narrative.
- Limited depth in result interpretation: Sections 2.6 and 3.1 mostly describe figures without discussing the implications of the results. Figure 4 is not intuitive and lacks numerical details to support the claims. A deeper discussion of what the observed trends reveal about social equity preferences or LLM alignment would make the results more meaningful.
- Missing critical methodological details: Section 2.4 (Question Data Generation) is overly brief despite being central to the paper’s contribution. It should describe in more detail how questions are generated for each (group, topic, dimension) combination. Similarly, the main evaluation metric (1 – Jensen–Shannon distance) is underexplained; the inputs, data scope, and evaluation setup should be clarified. The definition of “score” in the human evaluation is only understandable after reading the appendix. Key details should be moved into the main text.
- Presentation quality issues: The paper suffers from numerous presentation errors and clarity problems. Several figure references are incorrect (e.g., Figure 9 in the introduction should be Figure 2). Figure 1 is never discussed or explained. Descriptions in key sections (e.g., 2.1, 2.4, 3.2.3) are too brief to follow. The numbering-based labeling of segments and value dimensions significantly reduces readability, as readers must constantly refer back to definitions. Overall, the writing needs more coherence and care in exposition.

**Questions:**

- Could you provide more details about the human evaluation of the preference data, beyond what is described in Appendix G.4? Specifically, how many annotations were collected per question, how were they aggregated, and what was the inter-annotator agreement?
- What does the term “tier” represent in the method definition?
- Why was Segment 6 selected as the only alignment target? Were similar experiments conducted with other segments, and if so, why are only Segment 6 results reported?
- In Section 3.2.4, the paper notes that the prompt-based role-play method identifies models as Segment 6 in only 5 of 10 trials, which seems inconsistent with the 100% fidelity reported in Figure 5. Could you clarify the distinction between these two results?

---

### Note · Authors · 2026-01-06

I have read and agree with the venue's withdrawal policy on behalf of myself and my co-authors.